# An In Vitro Brain Tumour Model in Organotypic Slice Cultures Displaying Epileptiform Activity

**DOI:** 10.3390/brainsci13101451

**Published:** 2023-10-11

**Authors:** Harvey K. Chong, Ziang Ma, Kendrew Ka Chuon Wong, Andrew Morokoff, Chris French

**Affiliations:** 1Neural Dynamics Laboratory, Department of Medicine, University of Melbourne, Melbourne, VIC 3052, Australia; chong.harvey41@gmail.com (H.K.C.); kendrew.wong_health@outlook.com (K.K.C.W.); morokoff@unimelb.edu.au (A.M.); frenchc@unimelb.edu.au (C.F.); 2Department of Medicine, Royal Melbourne Hospital, Parkville, Melbourne, VIC 3000, Australia

**Keywords:** brain tumour, in vitro, organotypic, epilepsy, microvolume fluid sampling

## Abstract

Brain tumours have significant impacts on patients’ quality of life, and current treatments have limited effectiveness. To improve understanding of tumour development and explore new therapies, researchers rely on experimental models. However, reproducing tumour-associated epilepsy (TAE) in these models has been challenging. Existing models vary from cell lines to in vivo studies, but in vivo models are resource-intensive and often fail to mimic crucial features like seizures. In this study, we developed a technique in which normal rat organotypic brain tissue is implanted with an aggressive brain tumour. This method produces a focal invasive lesion that preserves neural responsiveness and exhibits epileptiform hyperexcitability. It allows for real-time imaging of tumour growth and invasion for up to four weeks and microvolume fluid sampling analysis of different regions, including the tumour, brain parenchyma, and peritumoral areas. The tumour cells expand and infiltrate the organotypic slice, resembling in vivo behaviour. Spontaneous seizure-like events occur in the tumour slice preparation and can be induced with stimulation or high extracellular potassium. Furthermore, we assess extracellular fluid composition in various regions of interest. This technique enables live cell confocal microscopy to record real-time tumour invasion properties, whilst maintaining neural excitability, generating field potentials, and epileptiform discharges, and provides a versatile preparation for the study of major clinical problems of tumour-associated epilepsy.

## 1. Introduction

Given the high mortality and morbidity of brain tumours and the lack of effective treatments, a variety of model systems have been developed, from in vitro tumour cell lines to in vivo animal models. A major goal of brain tumour research is to find a model that adequately recapitulates tumour-tissue interactions, as well as more complex effects such as the development of tumour-associated epilepsy, which occurs in a large proportion of subjects with considerable morbidity [1,2]. Current in vivo models for studying brain tumours consist of using genetic modification, transgenes, or knockout mice to induce tumour generation. These animal models have limitations, mostly lacking a strong similitude of tissue interaction, and it has been surprisingly difficult to observe spontaneous epileptic seizures. Although, seizure induction has previously been observed in a genetically modified immuno-deficient mouse model [3]. Other models include the use of transgenes or knockouts to induce tumour generation in experimental animals, and these mice can form tumours akin to human tumours. However, there remain a number of methodological concerns with these techniques, including off-target effects in non-tumour tissue of the gene modification, as well as potential confounding effects of viral delivery vectors [4].

As an alternative to in vivo approaches, we developed an in vitro model in which tumour cells with a well-defined lineage are implanted focally into rat organotypic brain slices [5]. This has the advantage of producing focal tumour growth and invasion in a well-defined neural matrix over substantial periods of time. It also preserves neural excitability and synaptic transmission, and the observation of seizure-like events allows for sampling of the tissue microenvironment and potentially functional testing of various drug effects and high-resolution real-time imaging.

## 2. Materials and Methods

The methodology of the in vitro brain tumour model described has several steps, which are outlined graphically in Figure 1. Further elaboration on each of the subsections of the methodology is provided in detail in the following text.

### 2.1. Ethics

All procedures were carried out in accordance with protocols approved by the University of Melbourne Animal Ethics Committee.

### 2.2. Brain Slice Harvesting and Culture

Six- to eight-day-old Wistar rats were rendered unconscious with isoflurane before decapitation and extraction of the intact brain into Gey’s Balanced Salt Solution (GBSS) supplemented with an additional 1 g/L of glucose. Hippocampal sections (350 µm thick) were cut with a McIlwain tissue chopper (Mickle Laboratory Engineering Co., Ltd., Guildford, Surrey, UK). Slices were cultured on 30 mm diameter tissue culture inserts (PICM03050, Merck Millipore, Burlington, MA, USA) as described by De Simoni and Yu (2006) [6]. It was also possible to place the slices on smaller pieces of PTE membrane (“confetti” [7]), allowing reuse of the membrane inserts. For the extended confocal live-cell imaging experiments, the slices were placed on poly-L-lysine-treated coverslips before positioning on the inserts. The slice culture media was composed of (in %*v*/*v*): 48%*v/v* Minimum Essential Media (MEM) + GlutaMAX (42360032, Gibco, Waltham, MA, USA), 24%*v/v* Earle’s Balanced Salt Solution (EBSS) without added calcium, magnesium and phenol red (14155, Gibco), 24 heat-inactivated horse serum (26050088, Gibco), 2%*v/v* of 32.5% *w*/*v* glucose dissolved in MEM, 2%*v/v* B27 supplement (17504044, Gibco), and 2%*v/v* penicillin–streptomycin (15070063, Gibco). All solutions were sterilised using 0.22 μm syringe filters (SLGP033RS, Merk Millipore). Slices were incubated at 37 °C with 5% CO_2_, 95% O_2_ gas, and the slice culture media was changed 24 h after harvest and every 48 h thereafter.

### 2.3. Cell Culture

SMA-560 cells, a spontaneous murine astrocytoma cell line [8], stably transfected with GFP, were provided by Dr. Hui Lau (Royal Melbourne Hospital) and cultured as an adherent monolayer in Dulbecco’s Modified Eagle’s Medium (DMEM, 10313021, Gibco) supplemented with (in %*v*/*v*) 10%*v/v* foetal bovine serum (12003C, SAFC Biosciences, Saint Louis, MO, USA), 1%*v/v* penicillin–streptomycin (15070063, Gibco) and 1%*v/v* GlutaMAX (35050061, Gibco) and incubated at 37 °C with 5% CO_2_, 95% O_2_.

### 2.4. Tumour Cell Injections

Tumour cells were injected into brain slices 3–5 days after preparation to allow adequate time for slice stabilisation and confirmation of viability.

SMA-560 cells were resuspended in Dulbecco’s Modified Eagle’s Medium (DMEM) at a density of 2 × 10^6^ cells/mL for injection into brain slices.

Borosilicate capillary glass (1 mm outer diameter, 0.58 mm inner diameter, 30-0017, Harvard Apparatus, Holliston, MA, USA) was used to make injection pipettes with a Sutter Instruments P-1000 electrode puller, after which the tip diameter was enlarged to facilitate cell delivery with gentle application of tissue paper. The final tip diameter was 300–500 μm.

Cell culture well inserts containing brain slices were transferred into a 35 mm culture dish with media to allow accurate positioning of the injection pipette. The insert was kept in the incubator until the injection apparatus was set up to minimize environmental exposure.

The injection pipette was inserted into the injector port of an IM-6 microinjector (Narishige) and positioned in the cell suspension. The suspension was carefully drawn up until approximately half the pipette was filled, being careful not to pull the cells into the attached tubing. A small amount of fluid was then expelled by slight positive pressure prior to positioning in the slice.

The filled pipette inserted into the microinjector was attached to a Narishige MM-3 micromanipulator for positioning in the slice. It was then guided just above the surface of the slice, to the point of injection, under a dissection microscope at 4× magnification. The tip was then lowered into the slice by approximately 100 μm. The cellular suspension generally started flowing at this stage due to loss of surface tension when contact was made with the slice; this was minimised by applying slight negative pressure to inhibit flow until the desired point of injection was reached. A bolus of ~1 μL was then injected. The extent of the labelled cell injection could be checked immediately under a fluorescent microscope. The “tumour load” could be adjusted by changing the volume of injection or cell suspension density, but generally, injection volumes less than 1 μL did not survive.

After injection, the insert was placed in the six-well plate and placed back in the incubator. The growth of the injected tumour mass could then be observed over several days (Figure 2C). The tumour mass tended to expand initially and then stabilise; the slices were typically used 10–21 days after cell injection for specific observations detailed below.

### 2.5. Electrophysiology

Brain slices adherent to the insert membrane were cut out and placed in a recording chamber ~2 mL volume with a coverslip bottom perfused at 1.5 mL/min with artificial cerebral spinal fluid (ACSF) (in mM: 120 NaCl, 4.5 KCl, 1 MgCl_2_, 2 CaCl_2_, 1.2 NaH_2_PO_4_, 23 NaHCO_3_, 11 D-glucose) bubbled with 5% CO_2_, 95% O_2_ gas. The present observations were made at ambient room temperature (~22 °C), but similar observations could be made at higher temperatures (32–34 °C) with a Warner TC-324C inline temperature controller. Glass pipettes for recording extracellular potentials were made from 1.5 mm thin-walled glass (30-0057, Harvard Apparatus), with tip resistances of ~4 MΩ and filled with ACSF. A tungsten concentric bipolar microelectrode (TM33CCINS, World Precision Instruments, Sarasota, FL, USA) was used for stimulation. The electrode was connected to an NPI SEC-05X amplifier and digitised at 10 kHz with an Analog devices PCI-6035E analogue to digital converter and low pass filtered at 2 kHz. A Grass SD9 stimulator provided stimulating pulses triggered from the A-D board. The software package WinWCP V5.5.6 (University of Strathclyde) was used to display and record signals.

“Field potential” recordings were acquired by stimulating Schaeffer collateral fibres in the stratum radiatum, with the recording electrode in the CA1 region. The stimulation voltage was then steadily increased until a maximal population spike was obtained. Stimulation was then adjusted to initiate primary after discharges (“PAD’s”) that resemble epileptic discharges and are sensitive to antiepileptic drugs, which are considered an in vitro model for evoked seizures [9]. PADs were activated with 1 s trains of 0.1 ms pulses at 100 Hz. The stimulation train amplitude was set at twice the stimulus amplitude needed to produce a field potential amplitude of greater than 0.5 mV.

In further experiments, epileptiform discharges were also evoked in both the control and tumour-bearing slices, respectively, using elevated KCl (8 mM) in the ACSF.

### 2.6. Extracellular Fluid Sampling

A modification of the method of Shao and Feldman (2007) was used to sample extracellular fluid from various areas of the brain slice, including the peri-tumoural region to measure for metabolites, neurotransmitters, or other substances of interest [10]. A glass electrode, similar to that used for tumour cell injection, was made with a large (~500 μm) orifice to allow aspiration of small samples of extracellular fluid. Extracellular fluid samples of 5–10 μL were collected typically at a rate of 1–2 μL/min with the same apparatus used for tumour cell injection for analysis using capillary electrophoresis or other methods.

### 2.7. Histology

Brain slices were preserved in 10% neutral buffered formalin (NBF) prior to staining and fixation. After paraffin embedding, 5 μm sections were cut and stained with haematoxylin and eosin for examination with light microscopy.

### 2.8. Imaging

The slices were imaged macroscopically at low power (Olympus XL Fluor 4×; Olympus, Tokyo, Japan) with a non-inverted fluorescence microscope (Olympus BX51; Olympus, Tokyo, Japan) with the culture plate cover in place. With the slices cut from the inserts, individual cells could be observed using conventional differential interference contrast (DIC) optics with a 40× water immersion lens (Olympus LUMPlanFLN 40×; Olympus, Tokyo, Japan). Additionally, it was possible to obtain more detailed observations of tumour properties, including spread into normal tissue, using a confocal laser microscope system (A1R, Nikon; Nikon, Tokyo, Japan) with a 488 nm laser and 10× Fluo/Na 0.3 objective for live cell imaging over several days. In this case, the slices were cultured on coverslips without the membrane inserts to allow visualisation of the slice and tumour growth through the bottom of the chamber.

## 3. Results

The GFP-labelled cell mass roughly doubled in size after 6–7 days. A typical seeding resulted in tumour mass areas (as judged using fluorescent imaging) of about 0.1–0.3 mm^2^ (Figure 2A–C). Variations in size, growth rate, and shape of tumour masses were observed. The haematoxylin and eosin staining of the tumours in the brain slices suggested patterns of growth and tissue invasion resembling higher-grade tumours (see Louis et al. (2016) [11]). The tumour mass consisted of tightly packed elongated cells with dark stained nuclei, as well as vacuoles in the peri-tumour area (Figure 2A,B). The expansion of the tumour cells and tissue invasion under confocal imaging is shown in Figure 3, showing both Z stacks and 3D reconstruction views (Figure 3A–C). Slice quality and tumour cell growth on the coverslip base appeared comparable to the slices maintained directly on the permeable inserts.

Electrical stimulation of the Schaffer collateral pathway (SCP) resulted in typical field potentials in the CA1 region. Tetanic stimulation of the SCP resulted in primary after-potentials (PADs) and seizure-like discharges. Typical field potentials and PADs resulting from tetanic stimulation in both the tumour-injected and control slices are shown in Figure 4A–D.

Using this protocol, the tumour-injected organotypic hippocampal slices were found to be hyperexcitable compared with the control slices as they exhibited much higher rates of PAD events following high-frequency electrical stimulation (Figure 4A: 105.67 ± 20.43 vs. 48.25 ± 9.43, mean ± standard error of the mean).

PAD frequency in tumour-injected slices was significantly higher (Figure 4G). PAD duration was generally longer in the tumour slices but did not reach statistical significance (Figure 4F). It is important to note that the quantification of PAD frequency was obtained by dividing the number of PAD events by the duration of activity. The frequency of PAD activity could vary, usually waning in frequency and occurring in bursts for some OHSs towards the end.

Levels of neurotransmitters and metabolites such as tryptophan, glycine, and ammonia could be measured from the microvolume fluid samples. Quantitative mass spectrometry was used to detect these constituents and quantify them using area under the curve (AOC) measurements, which allowed comparisons between control and tumour slices as well as intra-slice comparisons. AOC measurements (*n* = 5 for each region) from parenchyma and the peri-tumoral and intra-tumoral regions in the control and tumour slices are shown in Figure 5.

## 4. Discussion

The organotypic brain slice method is a widely used experimental model. Slices remain viable for weeks with electrophysiological function, and key cellular components such as microglia are preserved [12,13].

Previous reports describe several approaches to studying the interactions between tumour cells and brain tissue in vitro. Ren et al. (2015) [14] utilised rat organotypic slices as a substrate but used tumour cell aggregates rather than injection. Chuang et al. (2013) [15] described a co-culture system where tumour spheres are located next to the brain slice and allowed to invade the tissue. Jung et al. (2002) [16] used 1mm-thick white matter sections from resected human brain tissue with a cavity for tumour cell mass placement. Chadwick et al. (2015) [17] describe a method similar to the present study in that labelled tumour cells were cultured in mouse organotypic brain slices. However, there was a very diffuse (whole slice) infiltration of the tumour cells, and preparation viability was limited (~1 week).

More recently, the advanced in vivo and in vitro models such as the Hatcher and colleagues 2020 tumour suppressor gene deletion model [18] and Gill’s retroviral-induced animal model [19] have improved face and construct validity in the study of TAS. Gill’s retroviral-induced mouse glioma model allowed for the examination of the role of inhibitory interneuron in TAS generation [20].

The current report further extends these studies with the use of an intra-slice, volume-adjustable injection technique producing a focal invasive tumour resembling in vivo tumours. More significantly, it extends previous methods by permitting electrophysiological observations of neural circuit activation and epileptiform activity, live cell confocal imaging, and microenvironment sampling.

This model also avoids the need for genetic modification previously used in some in vivo models of brain tumour-associated epilepsy [3,18,21] and may be a potentially cost-effective platform for testing therapeutic interventions for brain neoplasia. The model may also be useful for exploring brain microenvironment interactions, which remains an area of considerable interest [22]. As the brain slices are clearly visible and easily accessible, they are open to a much wider range of manipulations than would usually be possible in vivo with an intact animal brain.

In principle, this technique may be performed with any form of fine injector such as a Hamilton syringe or even a volumetric pipette. However, the microinjector used in this study afforded excellent volume control and spillage prevention. Additionally, slices of any part of the brain could be used with this technique. The hippocampus was selected in this case due to an extensively studied structural connectivity that makes it particularly suitable for electrophysiological studies.

An interesting aspect of this method was the preservation of neural transmission, as well as the ability to generate epileptiform discharges. It is anticipated that this technique will be useful for studies investigating tumour-associated epilepsy pathogenesis and treatment. Electrophysiological measurements could potentially be extended with the use of multielectrode arrays (MEAs).

The ability to micro-sample extracellular fluid provides a range of further investigation. For example, it should be possible to test the “glutamate hypothesis”, which proposes that tumour-associated epilepsy is a result of excessive extracellular concentration of the potent excitatory neurotransmitter glutamic acid in the peri-tumoural brain [23]. Furthermore, assessment of the pathogenic role of other potentially significant tumour-related extracellular factors (e.g., exosomes) and signalling proteins (e.g., TGF-β) would be possible [24].

Aptamers are an emerging field in cancer diagnostics and treatment using single-stranded DNA or RNA oligomers, allowing for high binding specificity to cancer biomarkers [25]. Currently, aptamer labelling with ^99^mTc has been demonstrated to detect matrix metalloproteinases, a cell membrane-associated biomarker in glioblastomas [26]. The ^99^mTc labelling allowed for imaging of ex vivo human brain tumour slices. The high binding specificity and ease of modification of aptamers allow for their use in future studies for the investigation of tumour-related extracellular factors of interest.

The organotypic-based brain tumour model technique described in this article should also allow for genetic modification of the tumour or brain environment with AAV or lentiviral vectors. This, in combination with the ability to assess tumour-related extracellular factors in the model, increases the potential complexity of this tumour model, allowing for the study of genetic driver mutations’ influence on brain tumours. 

The ability to image at high resolution provides another significant advantage that may be useful for studying tumour invasion processes, including invadopodia activity and modulation in real-time. The technique may also provide a platform for testing brain tumour chemotherapeutic agents and other modalities of treatment, as has been shown by Chadwick et al. (2015) [17], but with a focal lesion.

Although SMA-560 with a GFP reporter was the only tumour cell line used, it is anticipated that other lines, including human-derived glioma cell lines, would be equally suitable. However, it is particularly useful to have the option of introducing a fluorescent label. Examples of other suitable cell types that could be used in this setting include “stem cell-like” tumour lines [27], which are currently an area of great interest.

Tumour cells of potentially any type may also be injected with this methodology as long as cell suspensions can be obtained. Other research areas that this technique could potentially be extended to include the exploration of metastatic tumour properties in other organs that can be cultured using the organotypic method.

## 5. Conclusions

This organotypic-based brain tumour model demonstrates rapid tumour cell expansion and invasion into a brain tissue substrate over relatively short periods of time, as imaged with GFP fluorescence, and mimics histologically the appearance of aggressively invasive tumour cell characteristics of the SMA-560 murine astrocytoma cell line. The slices are additionally useful for electrophysiological recordings, showing preserved electrophysiological responses and epileptic-like discharges. This technique may provide an alternative to in vivo models of brain tumour growth and a simple and less resource-intensive, high throughput platform for further studies.

## Figures and Tables

**Figure 1 brainsci-13-01451-f001:**
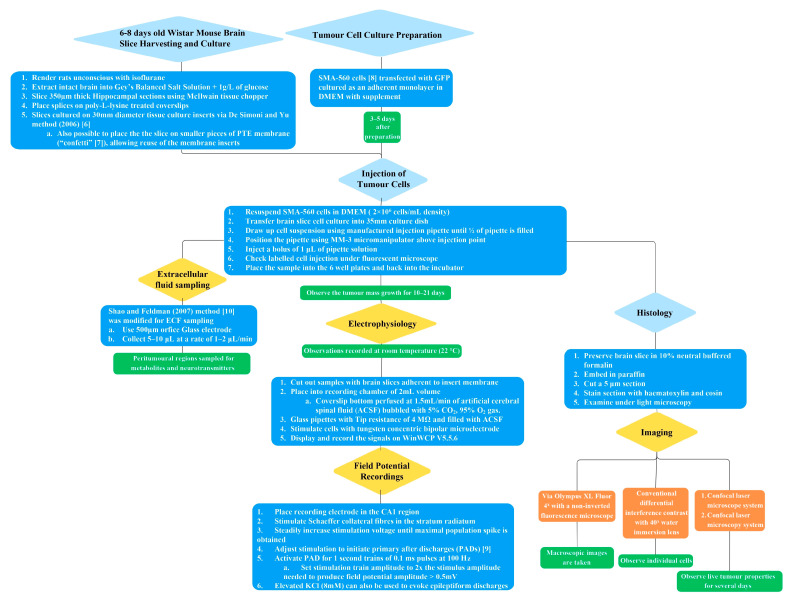
Flow diagram overview of the in vitro brain tumour methodology [6,7,8,9,10].

**Figure 2 brainsci-13-01451-f002:**
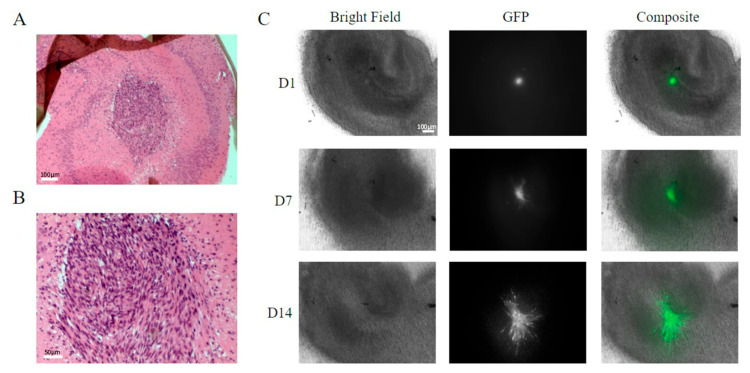
SMA560 tumour cells are able to grow and expand on organotypic hippocampal slices. Hematoxylin and eosin-stained histology sections of a tumour-bearing organotypic hippocampal slice 10 days post-injection under 4× (**A**) and 10× (**B**) objective magnification. Brightfield and epifluorescence micrographs of tumour-bearing organotypic hippocampal slices at select time points over 14 days under 4× objective magnification (**C**). Large, darkly stained nuclei form a dense central tumour mass extending into a normal brain with surrounding vacuoles at the borders similar to that seen with higher-grade human gliomas (**A**,**B**). Scale bar: 100 μm in (**A**) and 50 μm in (**B**). GFP-labelled SMA560 tumour cells can be seen spreading from the tumour origin at 1, 7, and 14 days post-injection (D1, D7, D14). Strands of tumour cells can be seen to emerge from the main tumour mass and form diffuse invasions with some preference for migrating around structures such as the dentate gyrus.

**Figure 3 brainsci-13-01451-f003:**
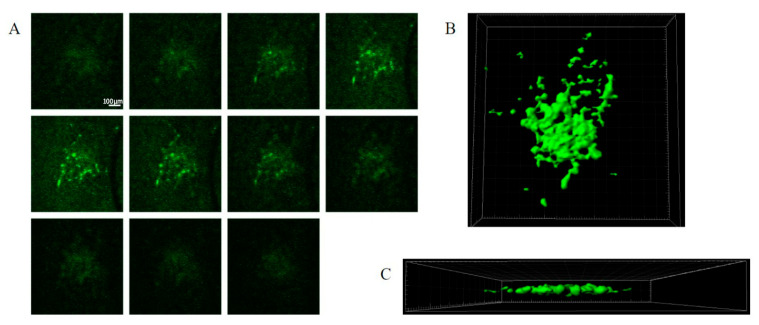
SMA560 tumour cells invade and infiltrate organotypic hippocampal slices. Z-series from live confocal fluorescence imaging of tumour-bearing organotypic hippocampal slices at 4 days post tumour injection (**A**). Iso surface renderings from live confocal fluorescence imaging of tumour-bearing organotypic hippocampal slices at 4 days post tumour injection in the X–Y (**B**) and X–Z (**C**) planes. Z-series progresses from left to right. Each image is 50 μm deeper along the z-axis (**A**). Scale bar: 100 μm in (**A**). SMA560 tumour cells are able to migrate in the z-axis, invading and infiltrating into the substance of the organotypic hippocampal slice. Most SMA560 tumour cells appear to migrate parallel to the slice surface with a few cells at the centre of the mass spreading into the z-axis (**C**).

**Figure 4 brainsci-13-01451-f004:**
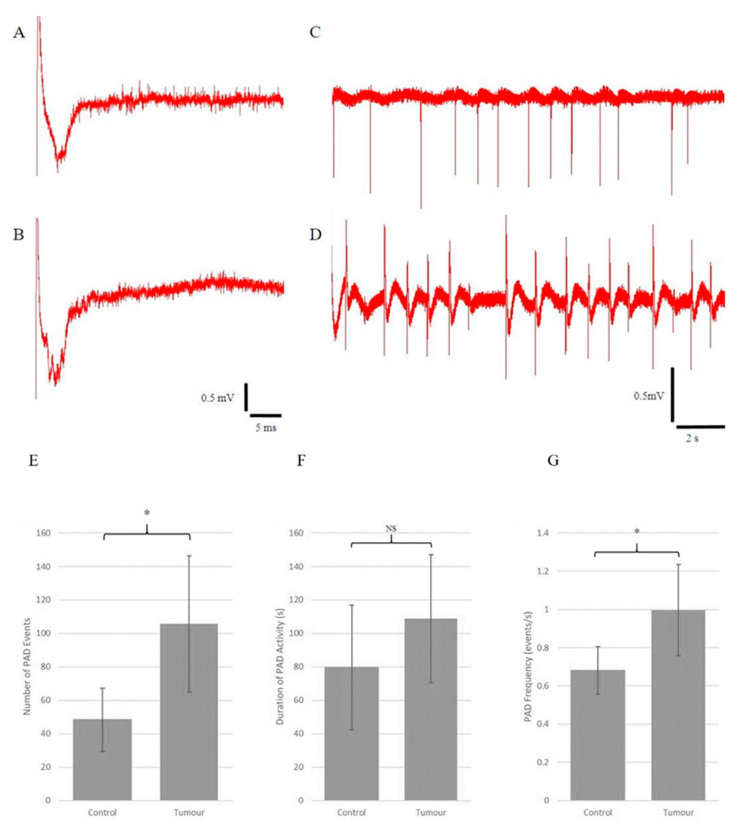
Organotypic hippocampal slices retain synaptic connections after SMA560 tumour cell injection and exhibit seizure-like primary after-discharges (PADs) and epileptiform discharges in a solution with 8 mM KCl or after high-frequency electrical stimulation. Recordings of evoked field potentials from control (**A**) and tumour-bearing (**B**) organotypic hippocampal slices. Representative examples of primary after-discharges from control (**C**) and tumour-bearing (**D**) organotypic hippocampal slices. Time and amplitude scales are provided for both recording groups at the bottom of the columns. Electrical activation of local field potentials shows that the tumour-bearing organotypic hippocampal slice has intact synaptic connectivity, most likely through the Schaeffer collateral pathway (**A**,**B**). (**E**) Tumour-bearing organotypic hippocampal slices exhibited statistically significantly more PAD events following high-frequency electrical stimulation (105.67 ± 20.43 events) compared with control organotypic hippocampal slices (48.25 ± 9.43 events). (**F**) No significant difference in the duration of PAD activity was found between tumour-bearing organotypic hippocampal slices (108.70 ± 19.09 s) and control organotypic hippocampal slices (79.57 ± 18.65 s). (**G**) Tumour-bearing organotypic hippocampal slices exhibited statistically significantly higher PAD frequency following high-frequency electrical stimulation (1.00 ± 0.12 events per second) compared with control organotypic hippocampal slices (0.68 ± 0.06 events per second). All data were normally distributed according to the Shapiro–Wilk test. Statistical analysis was conducted using Student’s *t*-test with α = 0.05. *n* = 8 for control organotypic hippocampal slices and *n* = 9 for tumour-bearing organotypic hippocampal slices. Data are presented as mean ± standard error of the mean * = *p* < 0.05. NS = not significant (*p* > 0.05).

**Figure 5 brainsci-13-01451-f005:**
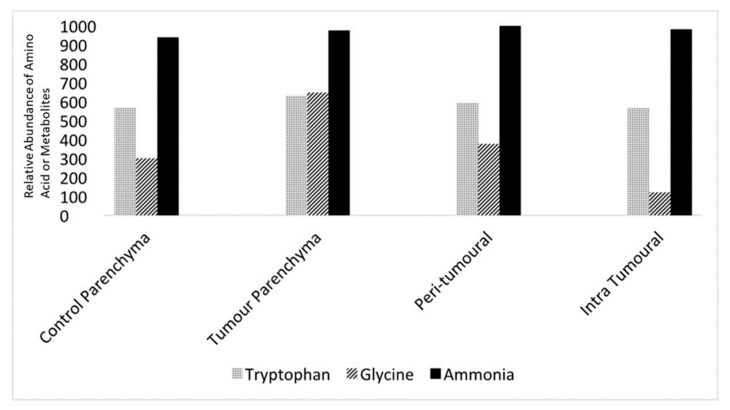
Micro-sampling of extracellular fluid reveals the relative abundance of amino acids and metabolites in different slice regions. Relative concentrations of tryptophan, glycine, and ammonia were derived from control brain slice parenchyma, tumour slice parenchyma, the peritumoral region, and the intra-tumoral region using 5–10 µL samples using the micro-sampling technique described above. Concentrations are relative areas under the curve derived from the quantitative mass spectroscopic technique using a battery of reference substances, *n* = 5 for all samples.

## Data Availability

Data is available upon reasonable request from corresponding author.

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
