# Peer review of "An In Vitro Brain Tumour Model in Organotypic Slice Cultures Displaying Epileptiform Activity"

_brainsci, 2023, doi:10.3390/brainsci13101451_

Round 1

Reviewer 1 Report

Oncogenic epilepsy is a serious problem in modern neurology and neurosurgery. The study of the mechanisms of epileptogenesis associated with the oncological process in the brain is undoubtedly important. The authors proposed an innovative technique that can significantly expand scientific knowledge not only in the formation of epilepsy, but also in the possibilities of drug treatment of this disease.

During the review, a number of comments and additions arose, which are rather of a technical nature:

1) The section "Materials and Methods" is described in detail, but for clarity it is recommended to present the course of the study in the form of a block diagram, with the addition of drawings or photographs for greater clarity and understanding of the methodology.

2) Figure 5 is not presented in full (text designations are cut off), it makes sense for the authors to replace this figure.

3) In terms of discussion, the question is interesting, have there been attempts to use aptamer-related technologies? For the purpose of labeling tumor structures and / or for the purpose of targeted delivery of antiepileptic drugs and chemotherapeutic drugs. If such research has not been done, is it possible in the future?

Author Response

Thank you for your detail feedback on our manuscript we have addressed some of the points you have brought up.

  • We have created a block diagram as you have suggested, this shows an overview of the methodology which is now titled Figure 1.
  • Thank you for pointing that out, we have reformatted the image now to show the full figure (now Figure 5)
  • There have been some attempts to use aptamer to label tumour structures and I have included this in the discussion now. Thank you for pointing that out, it could be a potential area of research applying it to our model.

Reviewer 2 Report

In this manuscript, Chong and colleagues developed a technique for implanting normal rat organotypic brain tissue into invasive brain tumors. This method produces focal invasive lesions that retain neural reactivity and exhibit epileptiform hyperexcitability. The tumor growth and invasion process are permitted to be imaged in real-time by this technique, allowing real-time tumor invasive properties to be documented using confocal microscopy. At the same time, in vitro culture of organotypic brain tissue preserves neural excitability and generates field potentials and epileptiform discharges, providing new ideas for research to address key clinical questions in tumor-associated epilepsy. Overall, this paper is an interesting study. However, I still have some concerns about the current form of the manuscript. The authors need to address several aspects before this can be published as follows:

Main concerns:

1. Figures 3 and 4 are representative plots of primary after-potential (PAD) events from control and tumor-bearing organotypic hippocampal slices and quantitative statistics, respectively, and should be combined into a single figure.

2. In Figure 3C, a representative example of primary after discharges of a hippocampal organotypic slice, there is only background noise.

3. Is Figure 3e from control or tumor-bearing organotypic hippocampal slices?

4. Do different periods of tumor invasion or tumor size have an effect on epileptiform discharges in hippocampal slices?

5. The quality of the pictures in Figures 3, 4, and 5 needs to be improved.

Moderate editing of English language required

Author Response

Thank you for your detailed feedback on our manuscript we have addressed some of the points you have brought up and provided further detailed explanations.

  • We have included a block flow diagram in the methods section to provide a brief overview of the methodology. Hopefully, this will provide greater clarity for the methodology.
  1. We agree the combination of Figure 3 and 4 (now Figure 4) is an improvement and we have deleted Fig 4e as it did not add useful information.
  2. Figure 3C (now 4C) shows control after discharges which I think is a useful comparison
  3. Regarding the tumour size, we did not attempt to correlate the tumour size with properties of the epileptiform discharges, but qualitatively there did not seem to be major differences except that electrical excitability appeared to attenuate with large tumour mass, most likely disrupting neural pathways. It would certainly be interesting in future studies to examine tumour load and epileptiform discharge properties.
  4. Improvement has been made in the quality of images overall

Round 2

Reviewer 2 Report

In this study, the author developed a technique for implanting normal rat organ type brain tissue into invasive brain tumors. This method produces a focal invasive lesion that maintains neural reactivity and exhibits epileptic like hyperexcitability. It can perform real-time imaging of tumor growth and invasion for up to four weeks, and perform microfluidic sampling analysis on different regions, including tumors, brain parenchyma, and surrounding areas.In the updated manuscript, the authors made improvements. These revisions basically answered my concerns. I have no more questions for the authors to answer.